# Mixed *Wolbachia* infections resolve rapidly during *in vitro* evolution

**Cade Mirchandani**[1,2], **Pingting Wang**[1], **Jodie Jacobs**[1,2], **Maximilian Genetti**[1,2], **Evan Pepper-Tunick**[3,4], **William T. Sullivan**[5], **Russell Corbett-Detig**[1,2], **Shelbi L. Russell**[1,2]*

**1** Department of Biomolecular Engineering, University of California Santa Cruz, Santa Cruz, California, United States of America, **2** Genomics Institute, University of California Santa Cruz, Santa Cruz, California, United States of America, **3** Institute for Systems Biology, Seattle, Washington, United States of America, **4** Molecular Engineering and Sciences Institute, University of Washington, Seattle, Washington, United States of America, **5** Department of Molecular, Cell, and Developmental Biology, University of California Santa Cruz, Santa Cruz, California, United States of America

\* shelbilrussell@gmail.com

**Data Availability Statement:** Short read sequencing data reported here have been deposited under the NCBI BioProject accession

## Abstract

The intracellular symbiont *Wolbachia pipientis* evolved after the divergence of arthropods and nematodes, but it reached high prevalence in many of these taxa through its abilities to infect new hosts and their germlines. Some strains exhibit long-term patterns of co-evolution with their hosts, while other strains are capable of switching hosts. This makes strain selection an important factor in symbiont-based biological control. However, little is known about the ecological and evolutionary interactions that occur when a promiscuous strain colonizes an infected host. Here, we study what occurs when two strains come into contact in host cells following horizontal transmission and infection. We focus on the faithful *w*Mel strain from *Drosophila melanogaster* and the promiscuous *w*Ri strain from *Drosophila simulans* using an *in vitro* cell culture system with multiple host cell types and combinatorial infection states. Mixing *D. melanogaster* cell lines stably infected with *w*Mel and *w*Ri revealed that wMel outcompetes *w*Ri quickly and reproducibly. Furthermore, *w*Mel was able to competitively exclude *w*Ri even from minuscule starting quantities, indicating that this is a nearly deterministic outcome, independent of the starting infection frequency. This competitive advantage was not exclusive to *w*Mel's native *D. melanogaster* cell background, as *w*Mel also outgrew *w*Ri in *D. simulans* cells. Overall, *w*Ri is less adept at *in vitro* growth and survival than *w*Mel and its *in vivo* state, revealing differences between the two strains in cellular and humoral regulation. These attributes may underlie the observed low rate of mixed infections in nature and the relatively rare rate of host-switching in most strains. Our *in vitro* experimental framework for estimating cellular growth dynamics of *Wolbachia* strains in different host species and cell types provides the first strategy for parameterizing endosymbiont and host cell biology at high resolution. This toolset will be crucial to our application of these bacteria as biological control agents in novel hosts and ecosystems.

PRJNA1091411. Code written and used in our analyses is available from https://github.com/shelbirussell/Mirchandani_et_al_2024.

**Funding:** This work was supported by UC Santa Cruz and the NIH (R00GM135583 to SLR; R35GM139595 to WTS; R35GM128932 to RCD; T32HG012344 to JJ). The funders had no role in the study design, data collection and analysis, decision to publish, or the preparation of the manuscript.

**Competing interests:** The authors have declared that no competing interests exist.

## Author summary

*Wolbachia pipientis* is one of the most common bacterial endosymbionts due to its ability to manipulate host reproduction, and it has become a useful biological control tool for mosquito populations. *Wolbachia* is passed from mother to offspring, however the bacterium can also "jump" to new hosts via horizontal transmission. When a *Wolbachia* strain successfully infects a new host, it may encounter a resident strain, possibly resulting in a superinfection of both strains, or replacement of the resident strain by the new strain. Here, we use a *Drosophila melanogaster* cell culture system to study the dynamics of mixed *Wolbachia* infections consisting of the high-fidelity *w*Mel and promiscuous *w*Ri strains. The *w*Mel strain consistently outcompetes the *w*Ri strain, regardless of *w*Mel's initial frequency in *D. melanogaster* cells. This competitive advantage is independent of host species. While both strains significantly impede host cell division, only the *w*Mel strain is able to rapidly expand into uninfected cells. Our results suggest that the *w*Ri strain is pathogenic in nature and a poor cellular symbiont, and it is retained in natural infections because cell lineages are not expendable or replaceable in development. These findings provide insights into mixed infection outcomes, which are crucial for the use of the bacteria in biological control.

## Introduction

The alphaproteobacterium *Wolbachia pipientis* became a widespread intracellular symbiont of arthropods and nematodes through its ability to infect novel hosts and establish germline transmission. Hundreds of millions of years after the divergence of Arthropoda and Nematoda (ca. 500 mya [1,2]), *Wolbachia* endosymbionts evolved (ca. 100–200 mya [3]) and spread to infect a high proportion of these hosts [4–6]. Following horizontal transmission to a new host and establishment of a stable infection, *Wolbachia* targets the host germline to achieve vertical transmission from one host generation to the next [4,7,8]. Thus, at least two core mechanisms have contributed to the rise of *Wolbachia* in ecdysozoan hosts: high infectivity and targeted germline transmission. These two traits appear primed for conflict, as natural selection for infectivity is often linked to pathogenicity, which could interfere with normal host development. However, they have harmonized in *Wolbachia* to produce the planet's largest pandemic [9].

 Significant variation exists among closely related *Wolbachia* strains in their ability to infect new hosts. While all strains examined undergo vertical transmission through the host germline [10], some strains are also adept at colonizing new hosts through horizontal transmission and novel infection establishment. Promiscuous *Wolbachia* strains, such as the *w*Ri strain from *Drosophila* [11] and *w*Jho from butterflies [12], are found in unrelated hosts or multiple hosts (*i.e*., superinfections, see S1 Fig). These strains often exhibit strong reproductive manipulations, such as cytoplasmic incompatibility (CI), that drive *Wolbachia* infections to high frequencies in host populations from low starting frequencies [13]. Indeed, recent biological control applications using *Wolbachia* infections rely on strong and predictable CI in non-native hosts for their spread across targeted populations [14]. Selection for beneficial host-symbiont emergent functions and phenotypes may also be sufficient to increase and maintain infection frequencies in strains lacking reproductive manipulations [4,15].

 Successful host-switches are the culmination of a successful horizontal transmission event, stable host colonization and propagation, co-option of germline transmission, and establishment across individuals in a population (S1 Fig and reviewed in [4]). Attempts to model *Wolbachia* infection distributions based on an average turnover process produce estimates that

explain global infection frequencies, but that fail to explain strain-to-strain variation in horizontal transmission ability and novel infection establishment [16]. A major challenge involves parameterizing the infrequent, but vital events in the process. Based upon the low rates of mixed infections in infected hosts and novel infections in uninfected hosts [17–19], joint rates of horizontal transmission and successful proliferation in a new host are exceedingly low. However, it is unclear whether both rates are low, or if horizontal transmission rates are high, but exceedingly few bacteria persist and colonize host tissues. Furthermore, it is unknown how divergent strains ecologically interact within a single host, especially if one strain is more promiscuous than the other.

To study the finescale ecological events that occur among endosymbionts and hosts in novel host infections, we developed an *in vitro Drosophila* cell line system infected with faithful and promiscuous strains of *Wolbachia*. We leveraged two different *Drosophila melanogaster* somatic cell types infected with the native *w*Mel strain and the non-native, promiscuous *w*Ri strain from *Drosophila simulans* to study what occurs when a promiscuous strain infects a host with a stable endosymbiont. Then, we use a novel *D. simulans* cell line immortalized for this study to explore the reciprocal mixed infection in one of *w*Ri's native hosts. Lastly, we measure infection expansion into uninfected host cells to parameterize a model of endosymbiont *in vitro* growth, host cell segregation, and cell-to-cell transfer. Overall, this work reveals that closely related strains have significantly different capacities for cellular proliferation that are counterintuitive based on their distributions among hosts. Furthermore, we show that mixed infections resolve rapidly and predictably across cell types and hosts, shedding light on the rarity of mixed infections in nature. These results significantly increase our understanding of what occurs when novel strains interact within host cells and tissues. This knowledge is critical to ensuring the safety of biological applications that release hosts infected with non-native *Wolbachia* strains into natural ecosystems.

## Results and discussion

### *In vitro Wolbachia* infections in *D. melanogaster* cell culture are stable over time

We successfully established and maintained *in vitro w*Mel and *w*Ri infections in two *D. melanogaster* cell lines, the neuroblast-like JW18 cell line [20] and the macrophage-like S2 cell line [21] (diagramed in S2 Fig). Fluorescence in situ hybridization (FISH) with 16S rRNA probes visually confirmed the presence of *Wolbachia* in infected cells (S1B–S1F Fig) and its absence from doxycycline-cured (DOX) cells (S3A–S3F Fig). We used whole genome sequencing (WGS) and reference genome mapping to confirm infection strain identities, estimate the genomic titer of each symbiont infected cell line, and observe fluctuations in titers over time. We consistently observe *w*Mel at a higher titer ($\sim$10–30) than *w*Ri ($\sim$0.1–3) (S3G Fig and S1 Table).

### The wMel strain outcompetes the wRi strain from equal starting ratios

The *w*Mel strain of *Wolbachia* outcompetes the *w*Ri strain in *D. melanogaster in vitro* infections. To recapitulate the conditions of a mixed *Wolbachia* strain infection *in vitro*, we mixed *w*Mel and *w*Ri infected cells at approximately equal genomic titers (Fig 1A). This equal starting ratio was selected to not advantage either strain and study the differences in infected host cell and strain growth rates. Each mixed culture was split into triplicate, and passaged every seven days, with a sample collected for sequencing at each passage. We estimated the abundance of each symbiont by calculating the proportion of total coverage contributed by that symbiont, which is the average coverage of the symbiont divided by the sum of the average coverages of

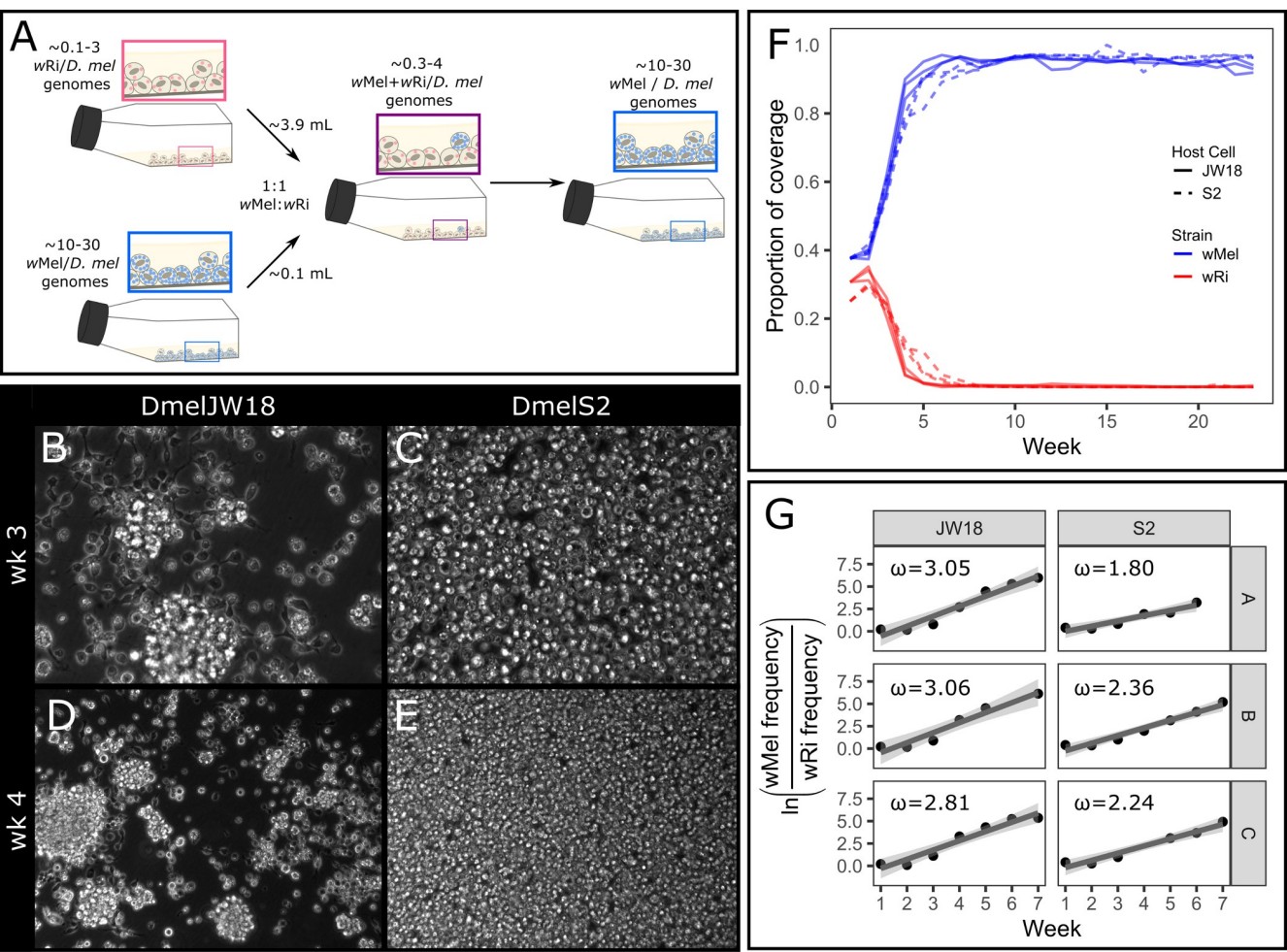

**Fig 1. The *w*Mel strain consistently outcompetes the *w*Ri strain in *D. melanogaster* cell culture.** A) Schematic overview of the 1:1 *w*Ri:*w*Mel mixed infected cell line experiment. B-E) Tissue culture micrographs of the mixed cell lines at B,D) week 3 and C,E) week 4. B,C at 40x and D,E at 20x magnification. F) Proportion of Illumina whole genome sequencing coverage mapped to the *w*Mel (blue) and *w*Ri (red) genomes out of the total coverage mapped to all *Wolbachia* and *D. melanogaster* host genomes, plotted by replicate and host cell type (S2, dashed or JW18, solid). G) Relative growth rates of *w*Mel compared to *w*Ri over the first seven weeks of *w*Mel exponential growth for the cell lines and replicates in (F). The slope of these plots was used to calculate the selection coefficients in S2 Table.

both symbionts and the host (S1 Table). In the first three weeks immediately following the initial mixing of the two strains, both *w*Mel and *w*Ri increased in frequency. However, after this phase of initial expansion, only *w*Mel continued to increase in frequency. By week five, *w*Mel accounted for an average proportion of total coverage of 90% (Fig 1F). During this timeframe, the JW18 neuroblast-like *D. melanogaster* cell culture cells exhibited adherence defects that suggested the cells were under stressful conditions, whereas the S2 cells maintained their normal phenotypes (Fig 1B–1E).

We used a simple haploid model of relative fitness (see Methods) to estimate the selection coefficient ($\omega$) of *w*Mel in the mixed infection experiments. Because we observed that *w*Mel replaces *w*Ri within five to seven weeks post mixing, we constrained our selection coefficient estimates to six weeks post-mixing in order to capture the early dynamics of selection acting on the two strains (Fig 1G). We estimated selection coefficients ranging from 2.81–3.06 in the JW18 cell line, and 1.80–2.36 in the S2 cell line (S1 Table). These values indicate that *w*Mel is

far fitter in *D. melanogaster* cells than *w*Ri. However, this selective advantage may have been influenced by *w*Mel's high starting concentration. Next, we explore whether *w*Mel outcompetes *w*Ri when it is a minority constituent in two-strain mixtures.

## Deterministic growth: wMel's selective advantage is independent of starting infection frequency

The *w*Mel strain outcompetes *w*Ri when it is the minority strain in host cell culture cells, indicating that *w*Mel is a deterministic competitor whose selective advantages are not dependent on starting infection frequency. To assess whether *w*Mel's competitive advantage is frequency-dependent or deterministic, we mixed *w*Mel-infected and *w*Ri-infected cells at approximately 1:100 and 1:1000 ratios based on the relative genomic titers of the respective strain in the stable-infected cell lines. The *w*Ri strain is at lower titer than the *w*Mel strain in both S2 and JW18 cells, limiting the titer mixtures to this value ($\sim$0.3–4.1). Relative titers were measured by Illumina whole genome sequencing each week over 11 weeks (S1 Table). Similar to the equal titer mixtures, we observed a rapid increase of the frequency of *w*Mel within five to seven weeks post-mixing in both the 1:100 and 1:1000 mixtures across both cell lines and all replicates (Fig 2A and 2B). However, in contrast to the 1:1 mixtures, the *w*Mel strain required more time to become fixed, only reaching an average proportion of total coverage of 86% by week 10 in both the 1:100 and 1:1000 mixtures.

The frequency of *w*Mel relative to *w*Ri increased continually over the 11 week experiment in both cell lines in the 1:100 mixtures, allowing us to estimate the strength of selection acting on *w*Mel over the total length of the experiment (Fig 2C). However, in the 1:1000 mixtures *w*Mel was undetectable in week 0, highlighting the extreme disadvantage in initial frequency when compared to *w*Ri. Therefore, we estimated selection coefficients for these mixtures from week one onwards (Fig 2D). In the 1:100 mixtures, selection coefficients (*ω)* for *w*Mel ranged between 2.62–2.92 and 2.31–2.33 in the JW18 and S2 cell lines, respectively. In the 1:1000 mixtures *ω* ranged from 3.36–3.63 in the JW18 cell line, and 2.76–2.87 in the S2 cell line (Fig 2F and 2G). Interestingly, we found that in the 1:1000 mixtures, the *w*Mel strain grows significantly faster than *w*Ri and exhibits higher selection coefficients than in the 1:100 mixtures across both cell lines (S4 Fig). This suggests that *w*Mel is able to modulate its growth rate to more efficiently populate host cells when starting at a lower initial frequency.

Given *Wolbachia's* propensity for recombination [22–24], we tested for the presence of recombinant haplotypes between the competing strain genomes in the 1:1, 1:100, and 1:1000 *w*Mel:*w*Ri mixed infection experiments. Putative recombinant events were detected by identifying overlapping chimeric alignments to both the *w*Mel and *w*Ri genomes in regions of high mappability. In total, we identified 67 recombinant events across all experiments. Although these events were extremely rare, we found they were relatively well distributed across both genomes. However, we did identify enrichment of recombinants in three 20kb regions in *w*Ri and two 20kb regions in *w*Mel (S5 Fig). The highest number of putative recombinant events occurred when strains co-occurred the longest, in the 1:1000 S2 mixtures. (S6 Table). These results make intuitive sense, as recombination mediated through passive processes such as homology-directed repair with divergent strain eDNA requires high concentrations (equal strain mixtures) and many chances (long co-culture times).

The competitive dynamics between *w*Mel and *w*Ri in our *in vitro* experiments offer insight into the mechanisms that might limit the frequency and stability of mixed infections *in vivo*, in nature. The quick and reproducible competitive exclusion of *w*Ri by *w*Mel in two *D. melanogaster* cell types across a range of starting frequencies suggests that mixed infections resolve reliably and quickly, consistent with theoretical predictions [25]. This potentially explains why

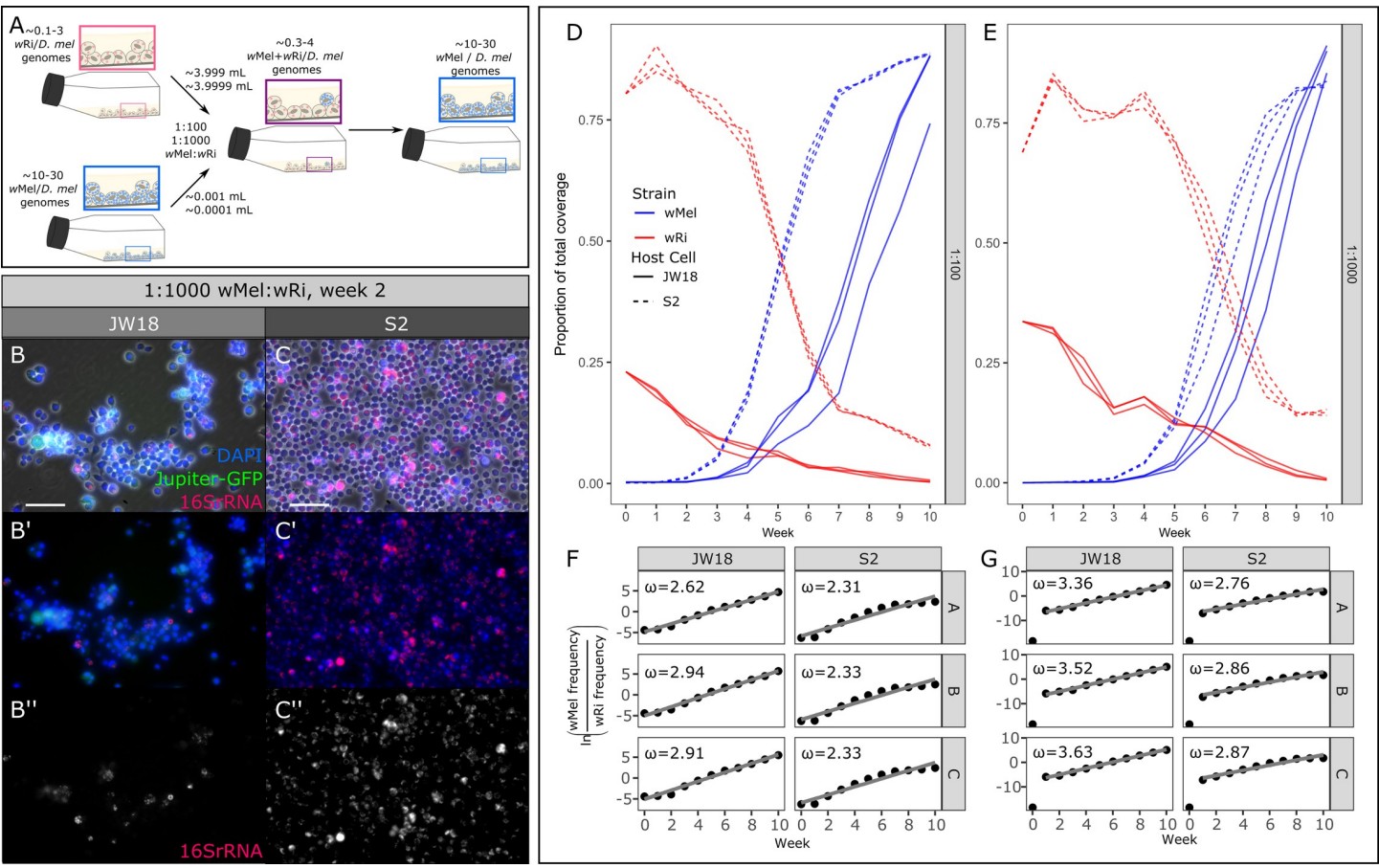

**Fig 2. The *w*Mel strain deterministically outcompetes the *w*Ri strain in mixed infections, even when starting at only 1/100th or 1/1000th the frequency of *w*Ri.** A) Schematic overview of the 1:100 and 1:1000 *w*Ri:*w*Mel mixed infected cell line experiments. B,C) Representative epifluorescence FISH images of week two of the 1:1000 *w*Mel:*w*Ri mixture (replicate A). B) JW18 cell line and C) S2 cell line at 20x; scale bar = 50 μm, DAPI = blue, Jupiter-GFP = green (JW18 only), and *Wolbachia* 16S rRNA = red. D,E) Proportion of *w*Mel (blue) and *w*Ri (red) genome coverage out of the total coverage of all *Wolbachia* strains and *D. melanogaster* host genomes, plotted by replicate and host cell type (S2, dashed or JW18, solid) in mixed infections started at *w*Mel:*w*Ri ratios of D) 1:100 and E) 1:1000. F,G) Relative growth rates of *w*Mel compared to *w*Ri in mixed infections started at F) 1:100 and G) 1:1000 ratios. The slopes from F and G were used to calculate the selection coefficients (ω) overlaid in the plots, also in S2 Table.

unstable mixed infections (opposed to stable superinfections) are rarely observed in nature [17–19]. The selection coefficients estimated for *w*Mel demonstrate a strong relative fitness compared to *w*Ri across both *D. melanogaster* cell lines. However, *w*Mel is natively associated with *D. melanogaster*, therefore this competitive advantage may reflect host-specific adaptations [26]. To explore whether the relative superiority of *w*Mel as a cellular symbiont is specific to its native host, we immortalized a *D. simulans* cell line to repeat these investigations in *w*Ri's native host background.

### Reciprocal infections: The wMel strain maintains its competitive advantage in wRi's native host *D. simulans*

To assess the contribution of host-specific adaptations to the competitive advantage of *w*Mel in *Drosophila melanogaster*, we immortalized a new *D. simulans* cell line from the white eye fly stock infected with the Riv84 *w*Ri strain named Dsim6B. Initially, these cells were heterogeneous and infected with *w*Ri (Fig 3A). Often the wRi-infected cells exhibited aberrant cellular morphologies. As the Dsim6B cell line stabilized and became more clonal, the infection was

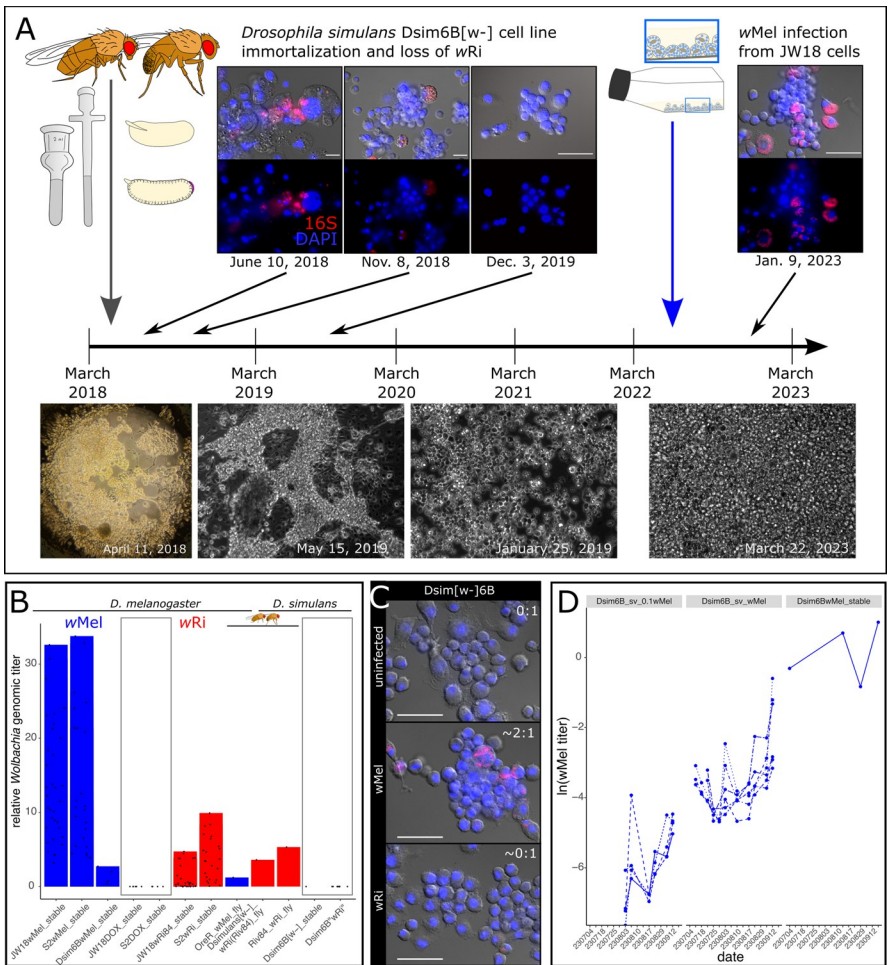

**Fig 3. The wMel strain is better at infecting *D. simulans* cells than *D. simulans'* native strain, wRi.** A) The Dsim6B cell line was immortalized from *D. simulans [w-]* embryos infected with the Riv84 strain of *w*Ri. The primary and early immortalized cell line was infected with *w*Ri, but the bacteria were gradually lost as the cells increased in growth rate and clonality. By nine months post-infection the Dsim6B cell line had cured itself of its *w*Ri infection. Repeated attempts to reinfect the Dsim6B cell line with *w*Ri were unsuccessful. B) Bar plots of stable *w*Mel (blue) and *w*Ri (red) titers in *D. melanogaster* and *D. simulans* cells and flies (three bars indicated with fly icons). C) FISH widefield images of Dsim6B cell lines uninfected (0:1 titer), infected with the *w*Mel strain (2:1 titer), and after attempts to reinfect with the *w*Ri strain (∼0:1). D) Titer increase over time in the Dsim6B cell line infected with *w*Mel via the shell vial technique at 1/10x the concentration in JW18 cells and at 1x, compared to stable Dsim6B*w*Mel cell line infections (maintained for more than three months).

lost (Figs 3A and S5). Despite high *w*Ri titers in *D. simulans in vivo* fly tissues (4.5x average genomic titer, Fig 3B) [6], repeated attempts to reinfect the cells with *w*Ri from the stably infected *D. melanogaster* cell lines via the shell vial technique failed (Figs 3C and S6). In contrast, infections of Dsim6B cells with the *w*Mel strain were very successful (S6 Fig), and the rate of titer increased to stable levels of 1-2x genomic titer depending on the initial input concentration (Fig 3D and S1 Table).

The differential success of *w*Ri and *w*Mel infections observed in our *D. simulans* cell line suggests that host developmental programs may enable the persistence of costly *Wolbachia* infections. Cell culture conditions are distinguished from *in vivo* conditions primarily by their simplicity of cell and organism types (sterile monoculture for both host and symbiont), which *w*Ri may be poorly evolved to handle, despite its close relationship to *w*Mel (99.91% identical

across the 1.3–1.4 Mb genomes). Alternatively, *w*Ri may be a better "developmental symbiont" than a "cellular symbiont". The *w*Ri strain's *in vivo* high titers and promiscuity across fly species suggests that its persistence may be heavily reliant on a developmentally-constrained system in which the maintenance of specific host cell lineages is crucial for organismal survival. In a cell culture system, cells can replicate freely because they are free of the limitations placed on cell proliferation in a developing host. Consequently, if the growth of uninfected cells outpaces infected cells, then the infection will be lost. Given that we were able to establish and maintain *w*Mel infections in both *D. melanogaster* and *D. simulans* cell lines, *w*Mel may not rely as heavily on the developmental context of the host as *w*Ri. To explore this idea further, we characterize the growth dynamics of each strain into uninfected host cells over time.

## Infection expansion into uninfected host cells recapitulates *w*Mel's spread into *w*Ri-containing cells

Successful *Wolbachia* cellular infections require healthy host cell growth, in addition to some rate of bacterial segregation during host cell division and cell-to-cell transfer to uninfected host cells. The weights of these three parameters are interdependent: if infections impact host cell growth, then cell-to-cell transfer rates need to be high to enable the infection of faster-growing uninfected cells. Otherwise, the infection will be lost due to uninfected cell overgrowth. Similarly, cell-to-cell transfer rates can only be negligible if the infection has minimal cost on host cell growth rates and segregation is efficient.

To understand the cellular basis for *w*Mel's competitive advantage over *w*Ri *in vitro*, we studied the expansion of these *Wolbachia* strains into uninfected host cells, revealing that *w*Ri fails to establish when fewer than 50% of host cells are infected. We mixed JW18 and S2 cells infected with the *w*Mel or the *w*Ri strain of *Wolbachia* and uninfected at approximately equal quantities. Infection growth curves following the addition of 1:1 uninfected host cells to *w*Mel-infected cell lines revealed a similar pattern of expansion as in the *w*Mel-*w*Ri competition experiments: across both the JW18 and S2 cell lines and all three replicates, *w*Mel genomic titer increased rapidly in the first five weeks, and remained at a relatively stable frequency throughout the rest of the experiment (Fig 4A and S7 and S1 Tables). On average, *w*Mel titer increased by 17% and 16% per week in the JW18 and S2 cell lines, respectively (Fig 4E and S3 Table). Conversely, in the *w*Ri-DOX mixtures, we observed the continuous decline of *w*Ri genomic titer in the JW18 cell line, with an average rate of 14% per week. Similarly, in the S2 cell line *w*Ri genomic titer declined on average by 10% per week, despite the initial increase in the first week post co-culture (Fig 4B, 4E and S3 Table). Overall, the observed patterns of *w*Mel's growth in the *w*Mel-DOX mixtures illustrate that the symbiont can effectively establish and expand an infection within the cell lines, and suggests horizontal transmission as a mechanism for infection establishment. To assess the impact of *Wolbachia* infection on host cell dynamics, we next compared the growth rates of infected and uninfected *D. melanogaster* cells.

Measuring the growth rate of *D. melanogaster* cells with and without *Wolbachia* infections revealed that both strains slow host cell division, suggesting that successful establishment requires cell-to-cell transfer. Both JW18 and S2 cell lines divide significantly faster when uninfected than when infected with either the *w*Mel or *w*Ri strain of *Wolbachia* ($p < 0.01$ Wilcoxon rank sum test; Fig 4C). When uninfected, JW18 cells double in 2.09 +/- 0.31 days (a growth rate of 1.45x cells per day), whereas *w*Mel-infected JW18 cells require 3.75 +/- 0.34 days to double (0.385x cells per day) and *w*Ri-infected JW18 cells require a massive 25.0 +/- 29.3 days to double (0.16x cells per day). Similarly, when uninfected, S2 cells double in 2.02 +/- 0.37 days (1.73x cells per day). When infected with the *w*Mel strain, S2 cells require 3.71 +/- 1.06 days to double (0.46x cells per day) and infected with *w*Ri, they require 3.21 +/- 0.63 days to double

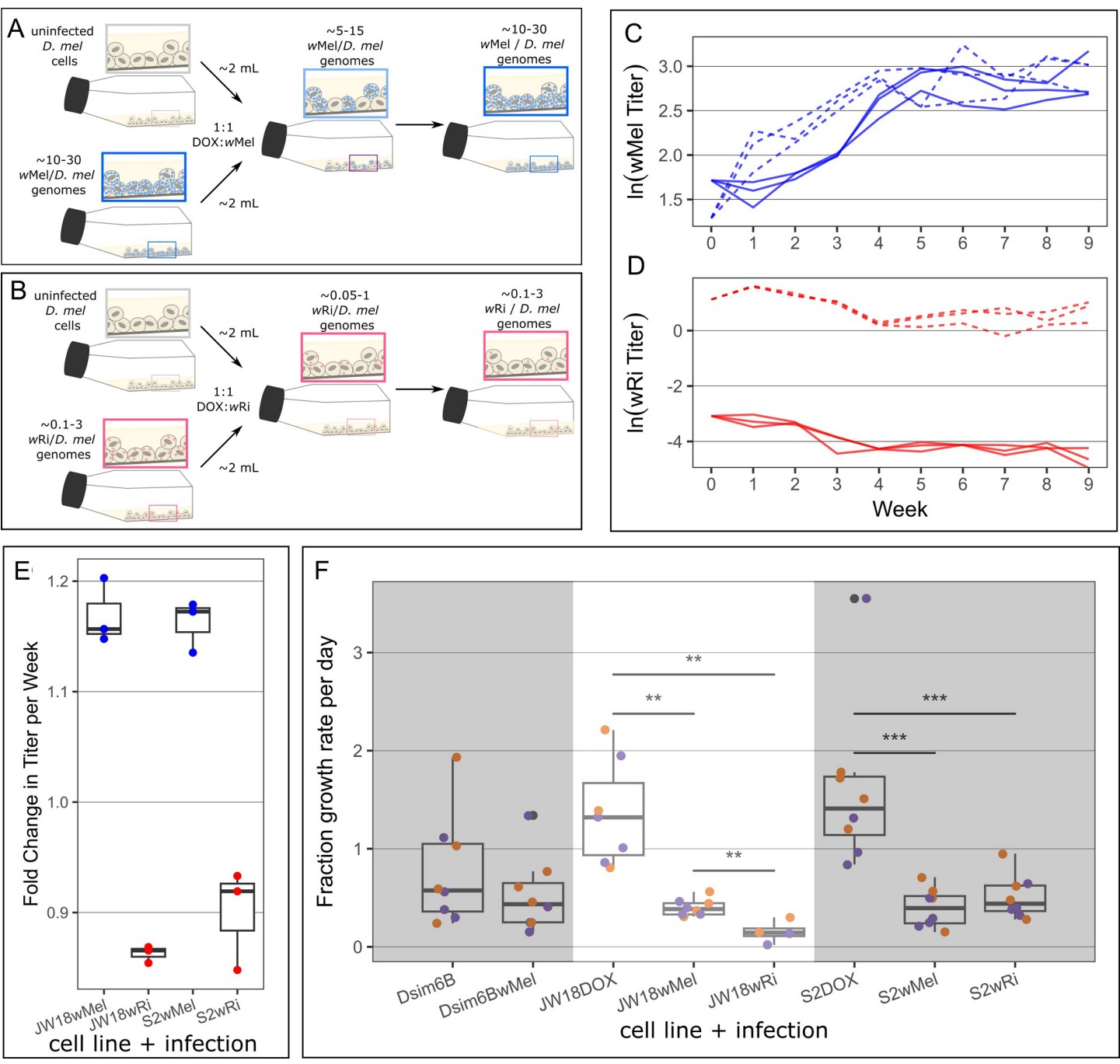

**Fig 4. The wMel strain is able to efficiently spread to uninfected cells through faithful segregation and cell-to-cell transfer, whereas the wRi strain cannot.** A) Schematic overview of the 1:1 *w*Mel:DOX and *w*Ri:DOX mixed infected cell line experiments. C,D) Genomic titers for C) *w*Mel (blue) and D) *w*Ri (red) over time in 1:1 mixtures with uninfected JW18 (solid line) and S2 (dashed line) cells. E) Fold change in symbiont titer per week in each mixture. Fold change was calculated by log-linear regression (S8 Fig and S3 Table). F) Cell growth rates measured by hemocytometer cell counts, quantified as the proportional growth per day from the starting cell count at 23˚C (purple) and 26˚C (orange). Wilcoxon rank sum p-values **p< = 0.01 and ***p< = 0.001.

(0.57x cells per day). Interestingly, *w*Mel *Wolbachia* infection has minimal impact on the *D. simulans* Dsim6B cell line (3.11 +/- 1.26 vs 3.23 +/- 0.95 days to double and 0.86x and 0.64x cells per day, respectively). This may be due to the Dsim6B cell line's lower growth rate: this

cell line is highly adherent and fails to grow well at the ⅙ starting dilution that the uninfected S2 and JW18 *D. melanogaster* cell lines thrive with.

The negative impact of *Wolbachia* infection on host cell growth combined with *w*Mel's ability to rapidly increase in titer upon exposure to uninfected host cells indicates that cell-to-cell transfer is essential to the colonization process. *D. melanogaster* host cells require nearly twice as long to divide when infected with *Wolbachia* than when uninfected (Fig 4C). The loss of *w*Ri from the 1:1 DOX-*w*Ri mixtures is consistent with the replacement of infected cells with faster-growing uninfected cells over the ten weeks of co-culture (Fig 4B). In contrast, *w*Mel's increase in frequency over time after 1:1 mixture with DOX-cured host cells (Fig 4A), despite their inhibition of host cell division rates (Fig 4C), is consistent with efficient cell-to-cell transfer to uninfected host cells. This transfer process not only increases *w*Mel frequency in the culture, but also prevents uninfected host cells from remaining uninfected and out-growing the infected cell population. Thus, despite not being able to specifically label, track, and distinguish living *Wolbachia* strains, we were able to detect indirect evidence of *w*Mel's superior cell-to-cell transfer ability, relative to *w*Ri.

The *in vitro Wolbachia* infections harbored low levels of genetic variation and this variation fluctuated across experimental time points. Average genome-wide pairwise diversity ranged from zero to 8.1e-05 (S9 Fig). Across experimental time points, diversity increased with increasing titer (S10 Fig) because detection of variation was limited by the depth of *Wolbachia* genome sequencing coverage (S1 Table). Despite the low levels of variation and the limits of sequencing depth in low-titer infections and time points, we did detect a handful of *w*Mel and *w*Ri SNP and indel alleles that fluctuate in frequency, but do not reach fixation (S11–S14 Figs). These alleles were shared among replicates and cell lines, suggesting that they were introduced as standing variation from the stable cell lines, opposed to being derived from *de novo* mutation in the experiments. The 15 high-frequency *w*Mel variant sites were located in hypothetical proteins, pseudogenes, ncRNAs, and protein-coding genes involved in oxidative stress and electron transfer (S7 Table). The five high frequency *w*Ri variant sites were located exclusively in pseudogenes and intergenic regions. While some of these alleles may confer fitness benefits *in vitro*, either for competition or infection, the mutations in intergenic regions and pseudogenes are likely neutral and pose no impact on the fitness of *Wolbachia*. These results hold promise for our ability to track allelic variation under natural selection in *Wolbachia in vitro* infections. Future work will confirm the functional nature of these mutations with deep long read sequencing.

## Conclusion

*Wolbachia pipientis* is an obligate intracellular alphaproteobacterium that infects a diverse range of arthropods, many of which are disease agents, vectors, and agricultural pests [10]. Composed of genetically distinct strains spanning 16 lineage groups [27], *Wolbachia* demonstrate a variety of interactions with their hosts, ranging from mutualism to reproductive parasitism [28]. The widespread prevalence of *Wolbachia* is largely due to its ability to rapidly shift to new and diverse hosts [4,29], but little is known about the microevolutionary events that occur immediately after a strain infects a novel host. Here, we used a *Wolbachia*-infected *Drosophila melanogaster* cell line system to investigate the outcomes of mixed and novel infections *in vitro*.

Our findings provide valuable insight into the ability of an invading *Wolbachia* strain to establish an infection in a host already infected by a different, resident *Wolbachia* strain. We show that *w*Mel consistently emerges as the dominant strain, quickly and effectively supplanting *w*Ri in mixed infections, independent of starting frequency. These results confirm

predictions made by Keeling et al. in 2003 [25], that one strain is always driven extinct in homogeneous mixed infections. However, the strain that wins is not determined by founder effects in the *w*Mel-vs-*w*Ri case, but the differential intrinsic abilities to propagate and colonize new host cells. These quick and reproducible resolutions of mixed infections in our cell culture system suggest an explanation for the paucity of observations from nature: mixed infections resolve quickly by competitive exclusion, before they can be sampled.

In addition to providing insight into *Wolbachia* infection establishment and mixed infection dynamics, this work highlights the potential role host development may play in determining the success or failure to establish an infection. Despite the promiscuous *w*Ri's strain's relatively high titer in whole-fly extracts (4.5x vs. *w*Mel at 0.79x, Fig 3B) and tissues [6,30], it occurs at titers an order of magnitude lower than *w*Mel in *D. melanogaster* cell lines (S2C and S2F Fig) and fails to persist in cell lines derived from its native *D. simulans* host (Fig 3C). This suggests that *w*Ri is costly at the cellular level and *in vivo* development offers a mechanism of protection from loss because most cell lineages are required for normal development. Similarly, *w*Mel's higher titer *in vitro* than *in vivo* suggests that host development and non-cell autonomous mechanisms are involved in their regulation in nature. Thus, in a developmentally constrained system, *w*Ri's high cellular cost and failure to transmit to uninfected cells (Fig 4B and 4C) do not prevent its persistence like they do *in vitro*.

The future of *Wolbachia*-mediated host biological control applications rely on understanding the mechanisms of novel *Wolbachia* infection and persistence in non-native hosts. From understanding which cell types and developmental time points different strains have affinities for, to predicting the outcome of rare mixed infections in unintended hosts, this work offers a powerful platform to disentangling bacterial-vs-host and cellular-vs-organismal driven phenotypes. Host *in vitro* systems could also provide a microcosm to select or engineer *Wolbachia* strains to be more permissive to new hosts and new cell type tropisms, expanding the utility of *Wolbachia* strains as biological control tools. Given that rare horizontal transmission events can produce mixed infections in novel hosts that may persist, generate recombinant *Wolbachia* strain genomes [5,31,32], and have unintended ecosystem-level consequences, these results are vital to future safe applications of *Wolbachia* in the field.

## Methods

### Cell culture maintenance and cell line generation

All *Drosophila* cells were maintained on either Shields and Sang M3 Insect Medium (MilliporeSigma S3652) or Schneider's Insect Medium (MilliporeSigma S0146) supplemented with 10% v/v Fetal Bovine Serum (FBS, ThermoFisher A3160502). Cells were maintained in 4 mL of media in plug-seal T-25 flasks (Corning 430168) in a refrigerated incubator at either 25–27˚C or 22.5–23.5˚C, as indicated in the text. We performed weekly cell splits at a 1:6 dilution for uninfected cell lines and 1:2 or 1:3 dilutions for *Wolbachia*-infected cell lines, following visual inspections of cell growth and contamination. Adherent cells were removed by scraping with sterile, bent glass pipettes. Transitions between media types were performed in 25% intervals, requiring four weeks to transition to 100% Shields and Sang or Schneider's Medium.

*Drosophila melanogaster* JW18 cells [20] and S2 cells (Thermo Fisher and [21]) were derived from a primary culture of 1–15 hr and 20–24 hr-old embryos, respectively. JW18 cells are naturally infected with the *w*Mel strain of *Wolbachia* (from the *in vivo* infection in the fly line the cells were derived from) and S2 cells are naturally uninfected. We found that incubation temperature exerts an observable effect on *Wolbachia* density within these cell lines, supporting the well established relationship between *Wolbachia* density and temperature [33]. Specifically, cells cultured at 26˚C in 2021 exhibited higher symbiont titers compared to the

same cultures incubated at 23˚C in 2023. Importantly, the relative differences between *w*Mel and *w*Ri titer are consistent between these temperature regimes: *w*Mel is always at an order of magnitude higher titer than *w*Ri. The difference in incubator temperatures was necessitated by the last author's starting her new lab and buying a new incubator capable of maintaining 23˚C. To generate uninfected JW18 cells, we treated JW18 *w*Mel-infected cells with 10 μg/mL doxycycline in supplemented Shields and Sang media.

We generated the *Drosophila simulans* Dsim[w-]6B cell line from a w[−] (white eye) fly line previously infected with the Riverside 1984 strain of *w*Ri *Wolbachia* [34,35] according to the method described in [36]. Briefly, 1–20 hr old embryos laid on grape-agar plates by *Wolbachia*-infected flies were collected, surface sterilized, homogenized, and plated in flasks on rich media containing 20% FBS and *Wolbachia*-resistant antibiotics, 60 and 100 μg/ml penicillin-streptomycin and 50 μg/ml gentamicin. During the next six months of maintenance, two of the initial twenty seed flasks converted into immortal tissue culture lines. The Dsim[w-]6B cell line was selected for further pursuit due to its planar growth pattern and ability to hold a *Wolbachia* infection. The native *w*Ri infection is unstable in *Drosophila in vitro* culture systems long-term, as described in the Results and Discussion sections, and the natural infection was lost naturally over the course of the first year of culture.

*Wolbachia* infections were introduced by adding 1.2 μm-filtered infected cell lysate to uninfected *D. melanogaster* JW18 and S2 and *D. simulans* Dsim[w-]6B cells. Infected cell lysate was either obtained from *Wolbachia*-infected cell cultures or fly embryos (collected on grape plates, as described in [36]. All *w*Mel infections were initialized with the *w*Mel strain that naturally immortalized in the JW18 cell line [20]. This wMel strain is known to lack the octomom region [37], which is linked to high titer in the wMelPop strain [38]. Infected cells were serially passed through 5 μM and 1.2 μM syringe filters to produce *Wolbachia*-containing cell lysate. The *w*Mel strain was applied directly (in 3 mL lysate) to uninfected S2 cells to produce the S2wMel cell line in 2017. To produce the wRi-infected cell lines and the Dsim[w-]6B cell lines, we applied 0.5–1 mL of *Wolbachia*-containing cell lysate to a monolayer of uninfected host cells in a flat-bottom shell vial and centrifuged the bacterial cells down onto the cell surface in a swinging bucket centrifuge at 2500 x g for 1 hr at 15˚C (*i.e.*, the shell vial technique [39]). We transferred these cells to T-12 flasks in a final volume of 2 mL for five days before scraping and transferring the cells to a T-25 flask with 2 mL of fresh media. These lines were maintained by weekly 1:2 "soft splits", which removed no media.

All cell lines were validated with DNA-based probes and whole genome sequencing after construction and continuously during maintenance and experimentation. Cell line infection status was continuously monitored by PCR and fluorescence *in situ* hybridization (FISH) of *Wolbachia*-specific markers. Primers for the *Wolbachia* Surface Protein (WSP) gene [35] were used to confirm the presence and absence of wMel and wRi strains in infected and uninfected cell lines, respectively. Sanger sequencing of the WSP amplicons was performed by Azenta to confirm the strain-specific amplicon sequences. Oligonucleotide DNA probes complementary to the *Wolbachia* 16S rRNA sequence were used in FISH experiments following the protocol in [40] to confirm infections and estimate per-cell *Wolbachia* titer. Whole genome shotgun sequencing was performed with Illumina sequencing (see below) to confirm host species and *Wolbachia* strain identities and test for contamination.

## Mixed cell line experiments

We used average genomic titer measurements from sequencing stable cell line infections to calculate how many cells of each infection and host cell type to mix for the desired mixture

ratios. Host cell concentrations were quantified with a hemocytometer manually or with a Millicell Digital Cell Imager.

For strains A and B at titers of $Y_A$ and $Y_B$ symbiont cells/host cell, within host cells growing at densities of $X_A$ and $X_B$ cells/mL, mixed at a ratio of A/B, in a final volume of 4 mL per cell culture flask:

Volume of host cell culture infected with strain A = $V_A$ = 4 mL/($X_A$/$X_B$ * $Y_A$/$Y_B$ * 1/(A/B) + 1)

Volume of host cell culture infected with strain B = $V_B$ = 4 mL - $V_A$

Samples were collected prior to mixing, immediately after mixing, and weekly when splitting infected cell cultures into new flasks at 1/2 dilutions. For each culture at each timepoint, one mL of scraped and mixed cell-containing media was transferred to a 1.5 mL Eppendorf tube, the cells were centrifuged at 16,000xg at 4–10˚C for 10 min, supernatant was discarded, and the cell pellet was snap-frozen and stored at -80˚C until DNA extraction. Pellets were processed for library prep within one month of sample collection.

## Shell vial experiments

To monitor how *Wolbachia* infections spread across uninfected host cells following introduction with the shell vial technique [39], we performed shell vial infections as described above for the creation of novel cell lines. Given the limited material at the start of these protocols ($\sim 2$ mL per experiment and $< 1$ million cells), we waited until the transfer step to T-25 flasks to take the first sample for Illumina sequencing and genomic titer quantification. Host cell-free *w*Mel *Wolbachia* lysate was either added at the full concentration derived from host cell lysis or a 1/10 dilution to approximate the lower concentrations exhibited by *w*Ri infections.

## Cell growth rate experiments

Cell line cells were quantified upon splitting and seeding into new flasks and after a week's incubation with a hemocytometer and Millicell Digital Cell Imager. While handling the cell lines as described above for "Cell Culture Maintenance", we added one extra mL of fresh media to each flask so that one mL could be removed for sampling cell concentration and relative *Wolbachia* genomic titer (as a final step in the splits). These one mL samples were then quantified by counting cells in a 10 uL volume (X number of cells ($>$100) measured per Y number of boxes ($>$1 if $<$100 cells/box) * W dilution factor (2 if diluted by ½) * 10,000 mL$^{-1}$ = Z number of cells/mL). The rest of the cell suspension was pelleted by centrifugation (as described above), snap-frozen, and stored at -20-80˚C until DNA extraction. This process was repeated one week later, except cells were resuspended by scraping prior to media removal so that the week's worth of growth could be quantified. Cells were then diluted as described above for normal maintenance. This modified step was repeated every other week for six weeks, at most frequent.

## Cell imaging and image analysis

Cell lines and experiments were continuously monitored with a tissue culture (TC) microscope and imaged with a monochromatic digital camera. Weekly, stable line and experiment cell splits were imaged on Zeiss Primovert TC microscope or a Leica DMi8 inverted microscope for confluency and contamination.

Infections were confirmed by fluorescence in situ hybridization (FISH) using DNA oligonucleotide probes complementary to the *Wolbachia* 16S ribosomal RNA sequence, following the protocol in White et al. 2017 [40]. Briefly, for each cell type, infection state, or experimental replicate, 1 mL of confluent cells were pipetted from a T-25 flask into a 6-well dish (Corning) one to three days before fixation. Upon confluency in the dish, cells were fixed in 8%

paraformaldehyde in 1x phosphate-buffered saline (PBS) for 15 min at room temperature (RT). Following two washes with 1x PBS, cells were treated with prehybridization buffer, consisting of 50% deionized formamide by volume, 4x saline sodium citrate (SSC), 0.5x Denhardt's solution, 0.1 M dithiothreitol (DTT), and 0.1% Tween 20 in deionized water, for one and a half hours. Following prehybridization, cells were incubated in hybridization buffer (prehybridization buffer without Tween 20) containing 500 nM *Wolbachia* W2 fluorescent DNA probe (5-CTTCTGTGAGTACCGTCATTATC-3) (Bioresearch Technologies) [41] at 37˚C overnight. Wet kimwipes were added to the dish to prevent dehydration. The next day, cells were washed three times with 1x SSC with 0.1% Tween 20 at RT quickly, at RT for 15 min, and at 42˚C for 30 min. Next, the cells were washed with 0.5x SSC at RT quickly, at 42˚C for 30 min, and at RT for 15 min. These stringent washes aimed to remove unbound FISH probes from the cells. Finally, cells were washed three times with 1x PBS at RT before either staining with 3uM DAPI (4',6-diamidino-2-phenylindole) in 1x PBS for 10 min or mounting in Vectashield fluorescent mounting medium with DAPI (Vector Laboratories).

FISH experiments were imaged on a Leica DM5500B widefield microscope or an inverted DMi8 equipped with LEDs for epifluorescence imaging. Raw Leica images were processed in Fiji [42] and analyzed in R [43].

## Whole genome resequencing and analysis

**DNA extraction.** Cell pellets were lysed and digested using lysis buffer (100mM NaCl, 50mM Tris-HCl pH 8, 1mM EDTA pH 8, 0.5% sodium dodecyl sulfate) and Proteinase K (NEB). Reactions were incubated at room temperature overnight. Genomic DNA was purified from cell lysates using SPRI beads and quantified using a Thermo Fisher Qubit fluorometer and Qubit dsDNA Broad Range assay kit.

**Tn5 Library Prep.** We generated short-read sequence libraries using a custom tagmentation protocol adapted from [44]. The full protocol is available on https://www.protocols.io/ [45]. Briefly, Tn5 Tagmentation reactions were prepared as follows: 10ng gDNA, 1uL Tn5-AR, 1uL Tn5-BR, 4uL TAPS-PEG 8000 and nuclease free water to final volume of 20uL. See S4 Table for Tn5-A, -B, and -R oligo sequences. Reactions were incubated at 55C for 8 minutes then killed by transferring to ice and adding 5uL 0.2% sodium dodecyl sulfate. Tagmentation product was amplified using the KAPA Biosystems HiFi polymerase kit and unique indexed primers. Pooled libraries were size selected using the Zymo Select-a-Size DNA Clean & Concentrator Kit and NEB Monarch Gel Extraction Kit. Library pools were then quantified using the Qubit dsDNA HS Assay Kit and the Agilent TapeStation.

**Data processing.** We developed a Snakemake [46] workflow to estimate symbiont titers from the raw sequencing data (https://github.com/shelbirussell/Mirchandani_et_al_2024). First, we generated a composite reference genome consisting of the host and symbiont genomes (S5 Table). We then calculated per-base mappability scores across the merged genome using *genmap* [47] with the parameters "-k 150 -e 0". Next, reads are trimmed of sequencing adapters and filtered for quality using *fastp* [48]. We aligned the filtered reads to the composite reference genome using *bwa mem* [49]. The resulting alignments were then filtered using *samtools* [50], keeping only unique alignments with a mapping quality greater than 20. Additionally, we used *sambamba [51]* to mark optical duplicates in the filtered alignments. Next, we calculated the mean depth for each mappable (mappability = = 1) position in the merged genome using *mosdepth* [52]. Using mean depth statistics, we estimated symbiont

genomic titer using the following equation:

$$\text{Symbiont titer} = \text{mean depth of symbiont/mean depth of host}$$

Note, we only considered the 5 autosomal chromosomes (2L, 2R, 3L, 3R, and 4) of the host *Drosophila* genome for our titer calculations.

**Selection coefficient calculation.** We leveraged population genomics theory on selection between competing strains in a chemostat [53] to model selection in *Wolbachia*-infected *Drosophila* cell culture. Weekly splits with removal and disposal of the overlying media approximate a chemostat, as the number of cells is kept within a tolerance range and the physical and chemical resources are kept plentiful.

In a bacterial chemostat (extracellular or intracellular), the frequencies of strains *A* and *a* under selection at time t can be shown to be $p_t = (p_t\text{-}1)\omega_{11}/\dot{\omega}$ and $q_t = (q_t\text{-}1)\omega_{22}/\dot{\omega}$, respectively. Let the selection parameter $\omega = \omega_{11}/\omega_{22}$.

Measuring strain *A's* fitness as a fraction of strain *a's* fitness from one generation to the next is described by the equation $p_t/q_t = (p_{t\text{-}1}/q_{t\text{-}1})\omega$. Solving for any generation gives the formula, $p_t/q_t = (p_0/q_0)\omega^t$. The plot of $\ln(p_t/q_t)$ should be linear with a slope equal to $\ln\omega$: $\ln(p_t/q_t) = \ln(p_0/q_0) + t\ln\omega$.

Thus, the selection coefficient ($\omega$) for strain A versus strain a is given by the e raised to the slope of the line fit to the plot of relative strain frequency over time. In our calculations, p = frequency of *w*Mel and q = (1-p) = frequency of *w*Ri. Thus $\omega$ reflects *w*Mel's selection coefficient relative to *w*Ri. We fit linear regressions to the *w*Mel growth curves using R v4.1.2 [43] and ggplot2 v3.4.1 [54].

In order to understand the effects of cell line and starting infection ratio on relative strain frequency over time, and consequently selection coefficients, we used a linear mixed-effects model with autoregressive moving average using nlme v3.1–146 [55]. Then, we plotted the observed points, fitted lines, and 95% confidence intervals using ggplot2 v.3.4.1.

**Recombinant haplotype detection with Illumina sequencing.** To detect potential recombinant haplotypes stemming from recombination between *w*Mel and *w*Ri, we used a custom Python script to select paired-end alignments with mapping quality $> = 20$ where one end aligned with *w*Mel and the other *w*Ri. We then filtered these alignments further, removing any alignment and its mate if it had a supplementary mapping position. Next, we clustered the chimeric alignments across all timepoints within an experimental replicate. Clusters were generated by finding alignments that overlapped within a distance of (mean insert size + 5*insert stdev) bp, then we used bedtools [56] to only retain clusters that overlapped a mappable region in each genome. Finally, we identified clusters as putative recombination events if they were supported by at least two alignments in each sample where they were found. We then performed a permutation test to identify regions in each genome enriched with recombinant events. We plotted the count of putative recombinant events in 5kb windows as well as links between the *w*Mel and *w*Ri genomes using R v4.1.2 and Circlize v0.4.11 [57].

**Intra-sample genetic variation analysis.** We called within-host SNP and indel variants for the *w*Mel and *w*Ri *Wolbachia* strains in cell culture using the method from [58]. Briefly, we created pileup files from alignments for all samples from all experiments using SAMTools [50]. Then, we called variants and calculated pairwise diversity using the perl script from [58](, which only considers sites within one standard deviation of the average genome coverage, filters SNPs around indels, and requires an alternate allele count in excess of the cumulative binomial probability of sequencing error at that site. We computed the average genome-wide and 1000-bp window pairwise diversity for each *Wolbachia* genome and each sample with

custom perl scripts and plotted in R v4.1.2. Alleles across experiments were plotted by frequency with pheatmap (version 1.0.12; [59]) in R.

## Supporting information

**S1 Fig. Schematic overview of steps required for successful horizontal transmission.** Host-switching of an endosymbiont requires successful horizontal transmission, intracellular proliferation, germline targeting for vertical transmission, and a mechanism for population establishment. Here, we use an *in vitro Wolbachia*-infected cell culture system to study the early stages in this process (#1 and 2 in bold) that are often lost to chance. By focusing on closely related strains with promiscuous and stable host-associations, we can understand how cell identities, divergent hosts, and resident strains impact novel infection events.
(TIF)

**S2 Fig. Overview of *Drosophila in vivo* and *in vitro* resources.** The S2 and JW18 *D. melanogaster* cell lines were derived previously from fly embryos of unknown infection status and infected with *w*Mel, respectively. The Dsim6B cell line was derived in this work, from embryos from the *D. simulans* white eye fly line infected with the Riv84 *w*Ri strain (see methods panel through embryo homogenization). Uninfected cell lines were obtained by treatment with 10 µg/mL doxycycline (DOX) in the cell culture media for nine weeks, followed by at least two months recovery from antibiotic treatment mitochondrial effects. *Wolbachia* strains were swapped among cell lines with the shell vial technique (see methods panel through shell vial technique).
(TIF)

**S3 Fig. Natural and introduced *Wolbachia* infections in *D. melanogaster* cell lines are stable over time.** The *w*Mel strain is consistently at $\sim$10x higher titer than the *w*Ri strain in *D. melanogaster* cells. Titers measured in 2021 were from cells maintained at 25–26˚C, whereas titers measured in 2023 were from cells maintained at 23˚C. Temperature has a similar impact on both strains titers, with both exhibiting proportionately lower titers at 23˚C than 25–26˚C.
(TIF)

**S4 Fig. Mixed Effects Regression analysis of relative strain frequency.** To assess how cell line and initial infection ratios influenced *w*Mel's competitive advantage over *w*Ri, we utilized a linear mixed-effects model incorporating these variables as fixed effects. Prediction lines and 95% confidence intervals from the model and observed points for the two cell lines A) JW18 and B) S2 at starting ratios 1:100 (red) and 1:1000 (blue).
(TIF)

**S5 Fig. Recombinants detected between the *w*Mel and *w*Ri strains in *D. melanogaster* cell culture.** Recombinant alignments were detected by extracting the reads chimerically mapped to the *w*Mel and *w*Ri genomes in regions of high mappability (containing SNPs, indels, or structural variation).
(TIF)

**S6 Fig. Loss of the native *w*Ri infection during Dsim[w-]6B cell line immortalization.**
(TIF)

**S7 Fig. Shell-vial reinfection of Dsim6B[w-] cell line.** *w*Mel (top) and *w*Ri (bottom) strains of *Wolbachia*.
(TIF)

**S8 Fig. Log-linear regression analysis of infected-uninfected mixtures.** Log-linear regression analysis for A) *w*Mel and B) *w*Ri in 1:1 mixtures with uninfected cells. Regression summary statistics are annotated in S3 Table.
(TIF)

**S9 Fig. Pairwise diversity in 1 kb non-overlapping windows along the *Wolbachia w*Mel and *w*Ri strain genomes at within-sample population levels.** Less *w*Ri allelic variation was detectable than for *w*Mel because of *w*Ri's relatively low titer in the JW18 cell line.
(TIF)

**S10 Fig. Average genome-wide pairwise diversity across experimental time points.**
(TIF)

**S11 Fig. Within-sample *w*Mel segregating alleles colored by their frequency within each replicate and time point from the S2 cell line 1:100 and 1:1K mixed *w*Mel:*w*Ri infection experiments.**
(TIF)

**S12 Fig. Within-sample *w*Mel segregating alleles colored by their frequency within each replicate and time point from the JW18 cell line 1:100 and 1:1K mixed *w*Mel:*w*Ri infection experiments.**
(TIF)

**S13 Fig. Within-sample *w*Ri segregating alleles colored by their frequency within each replicate and time point from the S2 cell line 1:100 and 1:1K mixed *w*Mel:*w*Ri infection experiments.** Variation in the *w*Ri genome was not detected in the 1:1 experimental mixtures or JW18 mixtures due to the lower titer and *w*Ri genome sequencing depth of those samples.
(TIF)

**S14 Fig. Within-sample *w*Mel and *w*Ri segregating alleles colored by their frequency within each replicate and time point from the S2 JW18 cell line 1:1 *w*Mel:DOX (left columns, blue heatmap) and *w*Ri:DOX (right column, red heatmap) infection experiments.**
(TIF)

**S1 Table. Sequencing and read mapping statistics.** Illumina sequencing and read mapping statistics for all cell pellets sampled in the paper. In this table, *Wolbachia* strain titer is symb_A_mean_depth / host_mean_depth, opposed to the coverage-based titer presented in the rest of the manuscript, which is symb_A_mean_depth / sum(mean depth of host + symbs). Coverage-based titer can be calculated from the genome coverage values in the table.
(XLSX)

**S2 Table. Selection coefficients estimated in competition experiments.**
(XLSX)

**S3 Table. Regression statistics from log-linear regression analysis of *w*Mel:DOX and *w*Ri:DOX experiments.**
(XLSX)

**S4 Table. Oligonucleotide sequences used for Tn5 based library preps.**
(XLSX)

**S5 Table. NCBI RefSeq genome accessions for reference genomes used in bioinformatics analyses.**
(XLSX)

**S6 Table. Putative recombinant event counts.**
(XLSX)

**S7 Table. Within-sample *Wolbachia* alleles.**
(XLSX)

## Acknowledgments

We thank the UCSC Life Sciences Microscopy Center (RRID:SCR_021135) and Ben Abrams for training and the use of their microscopes. We thank Brandt Warecki, others in the Sullivan Lab, for their thoughtful comments and feedback.

## Author Contributions

**Conceptualization:** William T. Sullivan, Russell Corbett-Detig, Shelbi L. Russell.

**Data curation:** Cade Mirchandani, Pingting Wang, Jodie Jacobs, Maximilian Genetti, Evan Pepper-Tunick, Shelbi L. Russell.

**Formal analysis:** Cade Mirchandani, Shelbi L. Russell.

**Funding acquisition:** Shelbi L. Russell.

**Investigation:** Cade Mirchandani, Pingting Wang, Jodie Jacobs, Maximilian Genetti, Evan Pepper-Tunick, Shelbi L. Russell.

**Methodology:** Cade Mirchandani, Shelbi L. Russell.

**Project administration:** Cade Mirchandani, Shelbi L. Russell.

**Resources:** Shelbi L. Russell.

**Software:** Cade Mirchandani, Shelbi L. Russell.

**Supervision:** Shelbi L. Russell.

**Validation:** Shelbi L. Russell.

**Visualization:** Cade Mirchandani, Shelbi L. Russell.

**Writing – original draft:** Cade Mirchandani, Shelbi L. Russell.

**Writing – review & editing:** Cade Mirchandani, William T. Sullivan, Russell Corbett-Detig, Shelbi L. Russell.

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
