## [Decision Letter · Decision Letter 0]

15 May 2024

Dear Mirchandani,

Thank you very much for submitting your manuscript "Mixed *Wolbachia* infections resolve rapidly during *in vitro* evolution" for consideration at PLOS Pathogens. As with all papers reviewed by the journal, your manuscript was reviewed by members of the editorial board and by several independent reviewers. The reviewers appreciated the attention to an important topic. Based on the reviews, we are likely to accept this manuscript for publication, providing that you modify the manuscript according to the review recommendations.

Please be sure to address all concerns Reviewer 1 in particular raises very valid concerns about over interpretation given that the work is in cell culture. 

Sincerely,

Elizabeth A McGraw, PhD

Academic Editor

PLOS Pathogens

Debra Bessen

Section Editor

PLOS Pathogens

Michael Malim

Editor-in-Chief

PLOS Pathogens

orcid.org/0000-0002-7699-2064

Reviewer Comments (if any, and for reference):

Reviewer's Responses to Questions

**Part I - Summary**

Reviewer #1: This work addresses an interesting question in the field - how is it that different Wolbachia strains proliferate within and between populations? What happens during co-infections? Co-infections do exist in the field and are often between unrelated Wolbachia strains (A and B type). This may be an artifact of detection (MLST differences) and it is likely based on recent work that Wolbachia are actually a cloud of genetic variants within a single host (see Chu et al https://onlinelibrary.wiley.com/doi/full/10.1111/1744-7917.12566).

Regardless, here the authors try and use cell culture to quantify a strain's ability to persist in a host population. There are some major caveats to this approach that I do not believe the authors fully appreciate given their interpretations. That said, I wouldn't throw out the baby with the bath water as there are some very interesting observations here (especially the conclusion that recombination between strains is frequent enough to observe in the lab - Figure S5).

Reviewer #2: Summary: The paper studies what happens when wMel and wRi strains infect the same cell culture. wMel outcompetes wRi quickly and consistently. Notably the authors contribute a novel cell culture line of D.sim. Overall they setup a model of wolbachia cell-to-cell invasion which is quite interesting. In general, the paper is well written. The experimenters do a good job of planning out the next most logical experiments and so the paper has good flow and makes complete sense. The analysis is sharpened by the selection coefficients. Overall, it seems the real importance of this study is in setting up a quantifiably assay that can dissect competitive wolbachia cell-to-cell invasion and colonization, which is an interesting tool. Long-term this research could pave the way toward dissecting mechanisms that might make wolbachia more permissive to new hosts, which could be a great biotechnological help/tool. This might be emphasized in the discussion.

**Part II – Major Issues: Key Experiments Required for Acceptance**

Reviewer #1: For the experiments detailed in all the figures, my concern is that each Wolbachia strain impacts cell growth differently under these conditions. The authors clearly measure this later on in the text and come to the conclusion that wRi massively impacts the replication rate of the host cell. It is therefore not surprising, nor particularly interesting, that the wMel infected cell population would take over.

I would've liked to have seen a difference in cell to cell spread. The authors assume that Wolbachia is spreading cell to cell but do not actually test for this. An appropriate test would be to use two different FISH probes and flow sort based on signal. This would help to quantify whether what they observe is spread vs. a dominant effect on host cell replication.

Reviewer #2: Major comments:

One really interesting piece of data here - is that the authors seem to be able to quantify the recombination rate of 1/500,000 genomes. This is the frequency of how often wMel and wRi are recombining in the co-culture. This is very interesting to me. Where did those recombinations occur and what information was exchanged? Was it large whole swaths of genome or small patches? What genes were contained in the swapped information? more work should be visually done to extract the information from that very valuable dataset. Are the recombinants random, or do the same regions keep recombining? Can this information be extracted and presented?

Is the strain being compared wMelPopcorn? which is known to be more infectious than wMel? Somewhere, there should be a discussion on the Jw-18 cell line, and it’s particular wolbachia infection, to clarify this point.

Were there any genomic differences that emerged post competition? For example, does the sequencing data show any snps or fixations of mutations that were selected for - after the competition assay – when compared to the reads at the start of the competition assay or before? Can this information be extracted and presented?

**Part III – Minor Issues: Editorial and Data Presentation Modifications**

Reviewer #1: Throughout, I would be very cautious about making interpretations about Wolbachias' behavior in whole animals - and populations - based on its behaviors in cell culture.

Writing/interpretation:

“revealing differences between cellular and humoral regulation” - this statement is quite broad and does not consider the fact that the in vitro cell environment is really quite weird and does not well represent any biological system super well. As the authors know, the cells are often multinucleate and since immortalized, have all sorts of chromosomal and cellular oddities.

I would also caution the authors about this statement below: “Our in vitro experimental framework for estimating cellular growth dynamics of Wolbachia strains in different host species, tissues, and cell types provides the first strategy for parameterizing endosymbiont and host cell biology at high resolution. This toolset will be crucial to our application of these bacteria as biological control agents in novel hosts and ecosystems.”

If the authors wanted to start to answer questions about tissue tropism and growth dynamics therein, these cell lines are not the best place to begin. There is a collection of DGRC cell lines from different Drosophila melanogaster tissue types but even then, they are not the same genetic background which make complicate interpretation.

In the abstract, the statement that Wolbachia evolved after the divergence of arthropods and nematodes is not supported by any citation.

I found the use of the words “faithful” and “promiscuous” to be distracting and frankly, off-putting - why personify?

Line 44 - "When a Wolbachia strain successfully infects a new host, it often encounters a resident strain that it must either replace or co-exist with as a superinfection." Is this really true? Since the ranges for infection are so broad and infections are not often fully penetrant, I am not sure I'd make this claim.

Reviewer #2: Minor comments:

Future ideas: transcriptomics of wolbachia during co-infections might provide insights on what genes help out compete.

I think I understand that the competition results are not specific to only d.mel hosts because in D.sim cell lines wRi is lost, whereas wMel can be maintained stably. Is that a correct read on the data?

Another interesting finding is that Wolbachia slows doubling time of D.mel cells. Didn't Frydman find and publish that Wolbachia increase stem cell proliferation? can you comment on these paradoxes, is it just a fluke of different cell types? https://pubmed.ncbi.nlm.nih.gov/22021671/

Some of the text in the figures is so small its quite difficult to read.

PLOS authors have the option to publish the peer review history of their article (what does this mean?). If published, this will include your full peer review and any attached files.

Reviewer #1: No

Reviewer #2: No

Figure Files:

Data Requirements:

Reproducibility:

References:

---

## [Editor Report · Decision Letter 1]

10 Jul 2024

Dear Mirchandani,

We are pleased to inform you that your manuscript 'Mixed *Wolbachia* infections resolve rapidly during *in vitro* evolution' has been provisionally accepted for publication in PLOS Pathogens.

Best regards,

Elizabeth A McGraw, PhD

Academic Editor

PLOS Pathogens

Debra Bessen

Section Editor

PLOS Pathogens

Michael Malim

Editor-in-Chief

PLOS Pathogens

orcid.org/0000-0002-7699-2064
---

## [Editor Report · Acceptance letter]

22 Jul 2024

Dear Mirchandani,

We are delighted to inform you that your manuscript, "Mixed *Wolbachia* infections resolve rapidly during *in vitro* evolution," has been formally accepted for publication in PLOS Pathogens.

Best regards,

Michael Malim

Editor-in-Chief

PLOS Pathogens

orcid.org/0000-0002-7699-2064